# DiffTED: One-shot Audio-driven TED Talk Video Generation with Diffusion-based Co-speech Gestures

## Abstract

*Audio-driven talking video generation has advanced significantly, but existing methods often depend on video-to-video translation techniques and traditional generative networks like GANs and they typically generate taking heads and co-speech gestures separately, leading to less coherent outputs. Furthermore, the gestures produced by these methods often appear overly smooth or subdued, lacking in diversity, and many gesture-centric approaches do not integrate talking head generation. To address these limitations, we introduce DiffTED, a new approach for one-shot audio-driven TED-style talking video generation from a single image. Specifically, we leverage a diffusion model to generate sequences of keypoints for a Thin-Plate Spline motion model, precisely controlling the avatar's animation while ensuring temporally coherent and diverse gestures. This innovative approach utilizes classifier-free guidance, empowering the gestures to flow naturally with the audio input without relying on pre-trained classifiers. Experiments demonstrate that DiffTED generates temporally coherent talking videos with diverse co-speech gestures.*

## 1. Introduction

Co-speech gestures are an integral part of human communication, and their importance has fueled the rise of co-speech gesture generation. Yet, despite numerous approaches [11, 12] for generating gestures and talking avatar videos, a critical gap remains: simultaneously producing realistic gestures and talking head video outputs.

Audio-driven gesture generation approaches often focus solely on the gesture and not with producing rendered video results, such as in [11]. Audio-driven gesture generation methods have used several different network structures, such as LSTMs [5, 14]. Recently, methods using diffusion models have been growing in popularity, where these models excel in gesture diversity and are able to leverage a variety of network structures to maintain temporal coherence [2, 31]. These methods, while able to produce compelling gestures, still leave the problem of transferring the gestures to images to produce videos or else are limited to the use with virtual avatars.

Additionally, gesture generation methods in 3D methods are able to work on the skeleton and thus the translation to video is non-trivial. Though, the skeleton offers several advantages to gesture generation, especially when not tasked with rendering a final video, such as not taking into consideration rigid constraints such as limb length. The methods can work with angles and direction vectors and then later apply predefined lengths to limbs to generate realistic-looking skeletal representations [14, 27, 31].

When rendering videos, some method of translating the pose or 3D body must be used but this is non-trivial, especially when considering texture. Methods in 2D can inherently use actual people/bodies and operate in image space and thus do not have to perform this transfer. However, without the third dimension depth ambiguity can become an issue. This means that body size or limb length can change from frame to frame and create unrealistic gestures.

Using skeletal motion in 2D can be one solution but ambiguous angles in the 2D still provide some challenges. 2D audio-driven video generation methods such as ANGIE [13] learn an unsupervised motion representation rather than the skeleton but it is limited to the front-facing upper torso of the body and has a complex network structure requiring large amounts of data and long training times.

In this paper, we propose *DiffTED*, the first one-shot audio-driven TED-style talking video generation from a single image with diffusion-generated co-speech gestures. The existing methods [5, 12] rely on video-to-video translation [8, 23] to render end results and as such, are unable to make a one-shot video generation pipeline. We choose to create a one-shot video generation method to be able to create videos of an arbitrary person with an arbitrary speech audio, rather than be bounded by the training subjects or having to retrain for additional people. We propose instead to utilize another approach to facilitate the one-shot video generation, learned 2D keypoints of Thin-Plate Spline (TPS) mo-

Figure 1. **Overview of the proposed pipeline: DiffTED.** Given a source image and driving audio as input, we generate a gesture sequence, $x_0$, represented by TPS keypoints using the diffusion model. This sequence of TPS keypoints then serves as input into the video renderer to transform the source image and produce the final talking video with co-speech gestures.

tion model [30]. With the simple representation of 2D TPS keypoints we can utilize a diffusion model such as several of the 3D gesture generation methods. Diffusion models excel at generating diverse but coherent gesture sequences and maintain a relatively simple network structure. Additionally, the TPS keypoint representation provides a natural path to video generation [30]. Our method moves diffusion into the realm of gesture generation to generate learned 2D TPS keypoints driven by audio. The audio-driven TPS keypoints are then used to render each frame individually by transforming a single source image. The use of diffusion in 2D TPS keypoint generation method allows for the creation of compelling and diverse co-speech gestures that can be rendered into realistic videos. Our proposed DiffTED represents the first one-shot audio-driven co-speech gesture video generation method.

With DiffTED, we can render realistic talking videos with co-speech gestures from a single source image of an arbitrary person and a driving speech audio of arbitrary length, as demonstrated in the results provided in Sec. 4 as well as in the supplementary video. Additionally, the source code of this work will be released to the public upon paper publication.

The contributions of this paper could be summarized as:

- We propose DiffTED, the first framework that can achieve one-shot audio-driven TED-style talking video generation with co-speech gestures. Our framework is built on top of the TPS motion model in order to transform the single input image with the guidance of co-speech gestures represented with 2D TPS keypoints.
- We introduce a diffusion-based method for the generation of 2D TPS keypoints representing co-speech gestures. We demonstrate that the diffusion method performs better than the traditional LSTM-based and CNN-based

models for the purpose of TPS-warped video generation with co-speech gestures.

## 2. Related Works

### 2.1. Talking Video Generation

Existing works [5, 12, 13, 16, 28] synthesize talking video from a sequence of 2D skeletons [5, 16] or 3D models [12] with the rendering process being disjoint from the generation of the gestures. In Speech2Gesture [5] and Speech2Video [12] they generate the gestures using a GAN, however, their methods suffer from a lack of diversity due to problems inherent in GANs like mode collapse. Qian et al. [16] use a VAE to model the distribution of gestures by learning a template vector that is mapped to a gesture sequence. In ANGIE [13], they use an unsupervised motion representation instead of a human skeleton or model to help improve image fidelity in generation. In our work, we opt to use the learned 2D keypoints of the Thin-plate Spline (TPS) motion model [30] as a target for generation and leverage the TPS motion model to render the keypoints into images. Learned 2D TPS keypoints have also previously shown good results for emotion-guided talking face generation [9]. Different from previous works, we focus on talking video generation with co-speech gestures.

### 2.2. Co-Speech Gesture Generation

Recent gesture generation techniques have shifted focus to data-driven methods that use deep neural networks to leverage large co-speech motion datasets to directly learn a mapping between speech and gestures. Current works use a mix of representations for the speaker with there being a mix of 2D and 3D representations. Most works use a partial 3D skeleton of the upper human body sometimes including the

hands or face. There have been many approaches to the design of the co-speech gesture predicting DNNs focusing on input modality (text or audio) or architecture. Some works use the speech text, audio, or both as input [10, 27] and they may additionally include other contexts like speaker identity [27]. There have been many architectures used in co-speech generation with the use of transformers [13, 15, 19], RNNs [3, 12, 26], GANs [5, 12, 14, 27], VAEs [11, 16], flow-based models [1, 25] and recently diffusion models [4, 31]. There has also been the recent introduction of VQ-VAE [3, 19, 24] in works to help keep diversity in the generated gestures. In DiffGesture [31], they introduce the use of a DDPM-like model for co-speech gesture generation on the 3D keypoints of a partial 3D skeleton to try and solve the problem of generation of diverse gesture sequences. All these co-speech gesture generation methods do not pay attention to the problem of video generation after the gestures are generated. In this paper, we use a DDPM-like model on learned 2D TPS keypoints, which bridges the gap between co-speech gesture generation and one-shot video generation.

## 3. Method

In this section, we introduce our DiffTED. A framework overview is shown in Fig. 1. It consists of two main parts, video generation and a diffusion model for co-speech gesture generation. We first introduce the formulation of the problem, then discuss the video generation, and finally the diffusion model.

### 3.1. Problem Formulation

To accomplish one-shot talking video generation from a single image and a driving speech audio, we first collect video clips of $N$ frames and the corresponding speech audio $\mathbf{a} = \{\mathbf{a}_1, ..., \mathbf{a}_N\}$. We extract keypoints, $\mathbf{x} = \{\mathbf{p}_1, ..., \mathbf{p}_N\}$, from the image using a pre-trained keypoint detector from Thin-Plate Spline (TPS) motion model [30]. Keypoint sequences are normalized using the global mean, $\mu$, and standard deviation, $\sigma$. The normalized sequences are calculated as $\mathbf{x} = (\mathbf{x} - \mu)/\sigma$. Our gesture generation model generates the normalized keypoint sequence $\mathbf{x}$ conditioned on the audio sequence $\mathbf{a}$ and initial $M$ normalized keypoint frames $\{\mathbf{p}_1, ..., \mathbf{p}_M\}$. The model uses these $M$ keypoint frames to set the initial pose of the speaker and we also use them to interpolate between segments of longer sequences. For one-shot video generation, we take the keypoints from the source image to use as the initial keypoints. The generated keypoints are then used to drive the video generation.

### 3.2. Video Generation

For generating video frames, we use the Thin-Plate Spline Motion Model [30]. To do this, we make use of its dense

motion network and inpainting network. Since the keypoints used to train our diffusion model are from its keypoint detector, our generated keypoints maintain the same semantic meaning as expected by the dense motion and inpainting networks. We choose to omit the use of the background affine transformation because we generate the video from a single image rather than from a driving video at inference time. Each video frame is generated for the driving keypoint sequence separately based on the thin-plate spline (TPS) transformations between the keypoints from the input image and the current frame's keypoints. The dense motion network estimates the optical flow and occlusion masks which the inpainting network uses to generate the final image.

Each generated gesture sequence contains $N$ frames. Practically, this $N$ cannot be a large number and thus each sequence is limited in time. To generate longer gesture sequences and thus longer videos sequences must be stitched together. To connect two sequences the last $M$ frames of the first sequence are used as the initial $M$ frame input of the second sequence. The model does not perfectly predict the first $M$ frames to be the same as the contextual input, therefore the overlapping frames are interpolated. The final overlapping frames are thus defined as:

$$\mathbf{p}_i = \mathbf{p}_{prev,i} * \frac{M-i}{M+1} + \mathbf{p}_{next,i} * \frac{i+1}{M+1}, \qquad (1)$$

where $\mathbf{p}_{prev,i}$ and $\mathbf{p}_{next,i}$ are the $i$th frame of the overlap for the first and second sequences respectively, and $i \in \{0, ..., M-1\}$.

### 3.3. Diffusion-based TPS Keypoint Generation

Motivated by the success of recent diffusion models [7, 31], we propose a novel diffusion model-based approach for generating co-speech gesture keypoint sequences.

The goal of diffusion is given some data sample, $\mathbf{x}_0$, from the real data distribution $q(\mathbf{x}_0)$, to learn a model distribution, $p_\theta(\mathbf{x}_0)$, that approximates the real distribution.

The forward, or diffusion, process is a Markov chain, $q(\mathbf{x}_t|\mathbf{x}_{t-1})$ for $t = \{1, ..., T\}$ in which Gaussian noise, $\mathcal{N}(\mu, \sigma^2)$, following a variance schedule $\beta_1, ..., \beta_T$, is iteratively added to the data sample, $\mathbf{x}_0$, eventually leading to pure noise. This process is defined as:

$$q(\mathbf{x}_t|\mathbf{x}_{t-1}) = \mathcal{N}(\mathbf{x}_t; \sqrt{1-\beta_t}\mathbf{x}_{t-1}, \beta_t\mathbf{I}). \qquad (2)$$

The reverse, or denoising, process $p$, then goes in the opposite direction gradually taking away noise, to go from the pure noise back to the data sample. Since the reverse process is being trained to recover the data sample we also add the contextual information, $\mathbf{c}$ and define the process as:

$$p_\theta(\mathbf{x}_{t-1}|\mathbf{x}_t, c) = \mathcal{N}(\mathbf{x}_{t-1}; \mu_\theta(\mathbf{x}_t, t, \mathbf{c}), \beta_t\mathbf{I}), \qquad (3)$$

| Methods | FVD↓ | FID↓ | LPIPS↓ | Div↑ | BC↑ |
|---|---|---|---|---|---|
| GT | - | - | - | 68.79 | 0.8669 |
| EAMM [9] | 140.31 | 18.50 | **0.2049** | 60.75 | 0.8033 |
| S2G [5] | 155.53 | 23.37 | 0.2183 | 59.05 | 0.8540 |
| Ours | **64.35** | **11.64** | 0.2091 | **61.99** | **0.8660** |

Table 1. Quantitative comparison between our method (diffusion-based), EAMM, Speech2Gesture (S2G) methods, and the ground truth (GT).

where, the network predicts the mean $\mu_\theta(\cdot)$ based on $\mathbf{x}_t$, timestep $t$, and the context information $c$. Thus, we can start from Gaussian noise $\mathbf{x}_T \sim \mathcal{N}(\mathbf{0}, \mathbf{I})$ and iteratively take away noise to recover the data sample $\mathbf{x}_0$. In our case, the data sample to be recovered are the image keypoints of $N$ frames: $\mathbf{x}_0 = \{\mathbf{p}_1, ..., \mathbf{p}_N\}$.

For optimization of the network, we follow DDPM [7] in optimizing the variational lower bound on negative log-likelihood: $\mathbb{E}[-\log p_\theta(\mathbf{x}_0)] \leq \mathbb{E}_q[-\log \frac{p_\theta(\mathbf{x}_0)}{q(\mathbf{x}_{1:T}|\mathbf{x}_0)}]$ [7]. Eliminating constant items that do not require training and adding conditioning on the contextual information, $\mathbf{c}$, we rewrite the loss function to: $L_{noise}(\theta) = \mathbb{E}_q[\sum_{t=2}^{T} D_{KL}(q(\mathbf{x}_{t-1}|\mathbf{x}_t, \mathbf{x}_0)\|p_\theta(\mathbf{x}_{t-1}|\mathbf{x}_t, \mathbf{c}))]$. We further follow [7] to simplify the noise loss to:

$$L = \mathbb{E}[\|\epsilon - \epsilon_\theta(\mathbf{x}_t, \mathbf{c}, t)\|^2]. \tag{4}$$

Here, $\epsilon \sim \mathcal{N}(0, \mathbf{I})$ is Gaussian noise that the network, $\epsilon_\theta(\mathbf{x}_t, \mathbf{c}, t)$ is trying to predict. And with $\alpha_t = 1 - \beta_t$ and $\bar{\alpha}_t = \prod_{s=1}^{t} \alpha_s$, the noisy keypoint sequence $\mathbf{x}_t$, is defined as:

$$\mathbf{x}_t = \sqrt{\bar{\alpha}_t}\mathbf{x}_0 + \sqrt{1 - \bar{\alpha}_t}\epsilon_t. \tag{5}$$

Rather than training for all iterations of the diffusion process, training is done by uniformly sampling $t$, from between 1 and $T$. Additionally, the model is trained under both conditional and unconditional modes jointly.

Following DiffGesture [31], we adopt their implicit classifier-free guidance method of training. This involves jointly training conditional and unconditional models. The conditional model is conditioned with the contextual information, $\mathbf{c}$ and for the unconditional model, $\mathbf{c}$ is set to $\emptyset$. where $\mathbf{c}$ is the concatenation of the driving audio and the initial keypoints. The unconditional model is trained used with a probability of $p_{uncond} = 0.1$.

To generate a keypoint sequence with the trained diffusion model, we first start with Gaussian noise and then iteratively remove noise in $\mathbf{x}_t$. The network predicts both conditional and unconditional noises that are then scaled with parameter $s$:

$$\hat{\epsilon}_\theta = \epsilon_\theta(\mathbf{x}_t, t) + s(\epsilon_\theta(\mathbf{x}_t, \mathbf{c}, t) - \epsilon_\theta(\mathbf{x}_t, t)). \tag{6}$$

The value of the scaling parameter, $s$, can be increased or decreased to make a trade off between gesture diversity and quality. With a larger $s$, diversity will increase, but the generated gesture will reduce in quality. For the experiments discussed in Sec. 4, we use $s = 0.2$.

## 4. Experiments

### 4.1. Experimental Settings

**Dataset.** Our model is trained on the TED-talks dataset [18]. The training videos are downscaled to a resolution of $384 \times 384$, focusing on the upper part of the human body, and resampled to 25 FPS. Videos are in the range of 64 to 1024 frames. To train our model, we use the keypoints from the learned keypoint detector in [30] to get the ground truth keypoints for each frame. For video generation, the first image from each video clip is used.

**Metrics.** We use five quantitative metrics to evaluate our pipeline, three for measuring the final image quality and two for measuring only the gesture sequences.

- **Fréchet Inception Distance (FID) [6]**: Aims to measure the similarity between generated and real images, in an attempt to reflect the image quality as it would be perceived by humans.
- **Fréchet Video Distance (FVD) [22]**: An extension of FID to videos, assessing the overall quality of generated videos by evaluating temporal coherence and image quality.
- **Learned Perceptual Image Patch Similarity (LPIPS) [29]**: An attempt to evaluate perceptual similarity between images based on deep learning features, which corresponds well with human judgment.
- **Diversity (Div)**: To measure the diversity, we follow [31] and train an auto-encoder on the keypoints to extract features of the generated gesture sequences and measure the mean feature distance between generated gestures and the ground truth gestures.
- **Beat Consistency (BC)**: In order to determine how well the generated sequences align with the cadence of human speech, we measure the beat consistency as in [31], but as we do not have a skeletal structure, we instead use the change in velocity of keypoints in adjacent frames to detect motion beats.

**Implementation Details.** Because there is no existing method for one-shot video generation that can generate audio-driven co-speech gestures, we instead adapt two existing methods that generate 2D keypoints. The first method, EAMM [9] utilizes an LSTM-based architecture to learn a 2D keypoint detector. We then follow Speech2Gesture [5] to implement a 1D Unet [8, 17] to represent CNN-based models. In both EAMM and Speech2Gesture adaptations we train on our learned 2D keypoints rather than the face and skeletal keypoints from those two works. These keypoints are then used on the same TPS keypoint-driven image transformation framework.

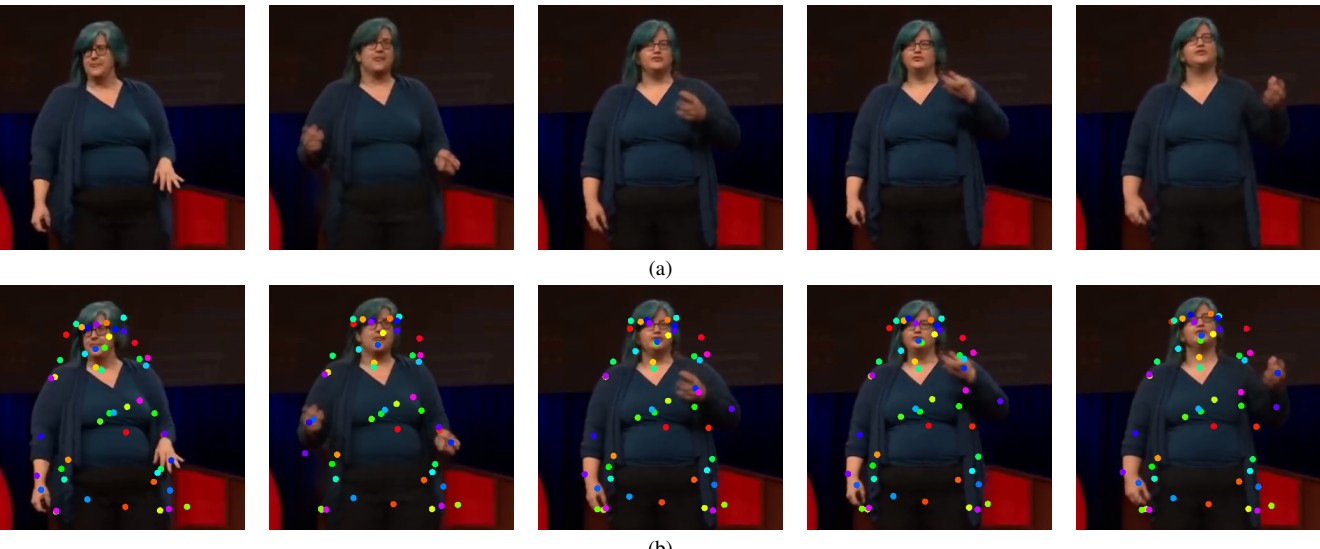

(a)

(b)

Figure 2. **Qualitative results of the DiffTED pipeline.** Five frames chosen from a sequence to show the diversity of gestures. The wide range of motion can be seen in the arms and the body positioning of the speaker, as well as in the direction the speaker is looking. In sequence (a) we can see movement in both hands as well as the face and body turning to look in a different direction. Sequence (b) is the same as (a) but with keypoints added.

| Methods | FVD↓ | FID↓ | LPIPS↓ | Div↑ | BC↑ |
|---|---|---|---|---|---|
| No Diff | 145.05 | 19.16 | 0.2059 | 60.75 | 0.8033 |
| Noise | **65.44** | **12.69** | 0.2116 | **61.99** | **0.8660** |
| Position | 103.64 | 16.61 | **0.1867** | 59.17 | 0.8633 |

Table 2. Ablation study. We show quantitative results for the method with no diffusion (EAMM-based method), diffusion on noise (ours), and diffusion on keypoint position.

For our training and testing, we use $N = 34$ frames with $M = 4$ frames of keypoints for contextual information. Audio processing is done as in DiffGesture [31] to get $N$ audio feature vectors of 32-D. In the training dataset, videos are sampled with a stride of 10 frames. In the testing set, the entire video is used and segmented into $N$ frame long clips with an overlap of $M$ frames. Only the first $M$ frames of the first clip are used as contextual information following the procedure discussed in Sec. 3.2.

For the diffusion model, we use timesteps of $T = 500$ and a linearly increasing variance schedule of $\beta_1 = 1e - 4$ to $\beta_T = 0.02$. The hidden dimension for the transformer blocks is set as 256 with 8 transformer blocks. We use an Adam optimizer with a learning rate of $5e - 4$. Training takes about 1 hour on an NVIDIA RTX A5000.

### 4.2. Experimental Results

**Quantitative Results.** Quantitative results with the five metrics between the diffusion model and the EAMM and Speech2Gesture models are shown in Tab. 1. The EAMM and Speech2Gesture methods show worse performance in both FVD and FID metrics, similar results for the LPIPS, and moderately worse performance in BC and diversity.

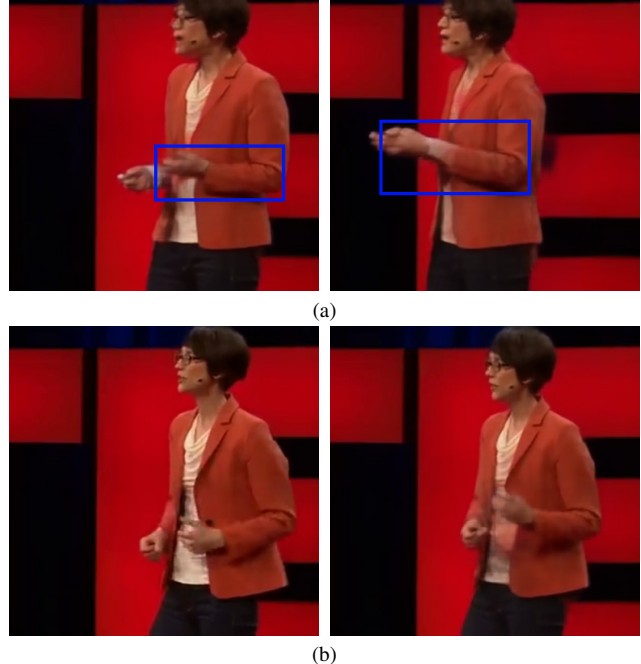

(a)

(b)

Figure 3. Failure case of the Speech2Gesture-based network where the arm, highlighted in blue, grows throughout the sequence in (a). Where in the diffusion network, the relative arm length in the sequence stays the same size as shown in (b).

Since the rendering method does not change between either the diffusion-based or the EAMM and Speech2Gesture models, the results compare the quality of the gesture generation of TPS keypoints. In both the EAMM and Speech2Gesture models, the FVD score is significantly

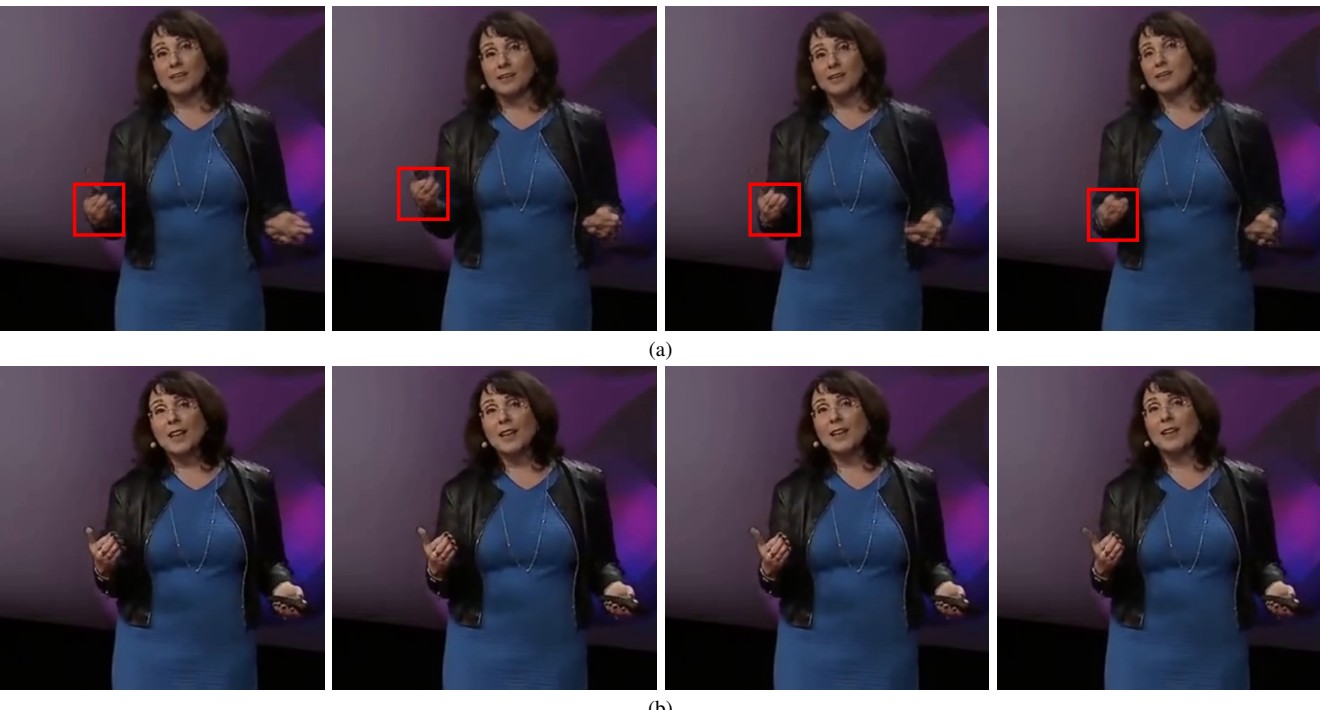

(a)

(b)

Figure 4. The EAMM-based method suffers from jittering effects in the generated gestures. (a) show 4 subsequent frames that have a quick jitter seen in the hand, highlighted in red. The hand moves from the initial position in the first frame to a raised position in second, back to the initial position in third, and then lower in the fourth. A smoother and more gradual transition between poses is expected as seen in the sequence of (b), which is generated by our diffusion-based method.

worse than the diffusion model. The FVD metric takes into consideration the temporal coherence of a video, where the EAMM and Speech2Gesture models trail behind our method.

**Qualitative Results.** In Fig. 2, we show several frames from a sequence to showcase gesture diversity. We show the sequence with (Fig. 2b) and without (Fig. 2a) the diffusion generated keypoints. The gestures shown have a wide range of motion in both the arms of the speaker.

Figure 3 provides a failure case for the Speech2Gesture model in which the speaker's arm grows in length showing that the model is unable to maintain consistent sizing of limbs. Maintaining limb size is an important aspect of creating realistic and believable videos of humans, the diffusion model is able to maintain believable transformations of the arms unlike in the Speech2Gesture model. Similarly, in Fig. 4, we show an example of a jittering motion that is common to sequences generated by the EAMM model. Smooth gestures and smooth transitions between poses seen in the diffusion model's output show that diffusion is able to create temporally coherent gestures, whereas the EAMM model struggles with always maintaining that coherency.

**Ablation Study.** We also perform an ablation study to compare the use of the pipeline with no diffusion, with diffusion on the noise, and with diffusion on keypoint position. The results of this ablation study are shown in Tab. 2. The non-diffusion method uses the EAMM-based network to produce keypoints. The diffusion on the noise is using the pipeline as described in Sec. 3 with the training objective Eq. (4). The diffusion on the keypoint position method is replacing the Eq. (4) with a loss on the keypoint position instead of the generated noise. The keypoint position loss is defined as:

$$L = \mathbb{E}[\|\mathbf{x} - \hat{\mathbf{x}}_\theta(\mathbf{z}_t, \mathbf{c}, t)\|^2]. \tag{7}$$

Here, instead of predicting the noise we directly diffuse the keypoint positions, $\hat{\mathbf{x}}_\theta(\cdot)$. The noise, in the base diffusion model is subsequently removed from the noisy sample, but in this method, the denoised sample is instead predicted directly. This method has been used recently in EDGE [21] and MDM [20]. In these works the method is shown to give better results and introduces the ability to add additional losses on the data sample directly. In our ablation study, this method does not perform as well as noise prediction and may require additional metrics to outperform the baseline diffusion model.

In Fig. 5 we illustrate one of the differences in results between diffusing on the position rather than on diffusing on the noise. The diffusion on position examples show an unnatural bend in the arms of the subject while diffusing

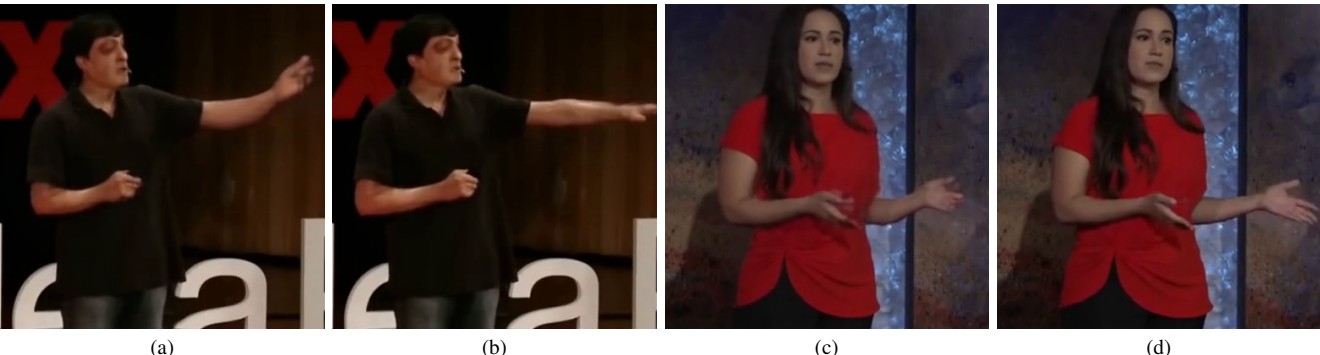

Figure 5. Qualitative example of ablation on diffusion on position (a)(c), and diffusion on noise (b)(d). In (a), the outstretched arm has an unnatural bend to it, while in (b) the arm is straight. Image (c) shows another example of an unnatural bend in the arm, where in (d) the arm is straight as expected.

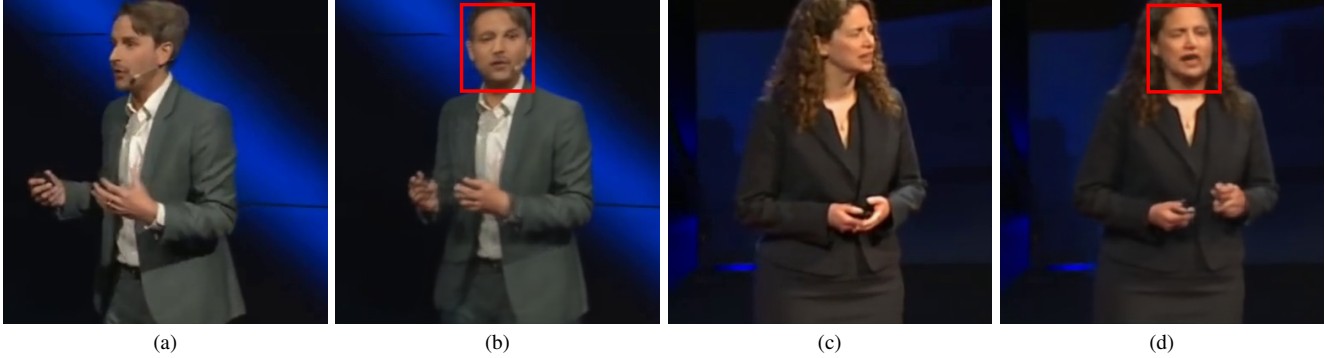

Figure 6. Example of distorted face artifact if the starting image is facing one side. With (a) as the source image of the video, when the person turns to face forward the face of the subject will be distorted as seen in (b). Image (c) shows another example of a source image with a person looking to the side and (d) the resulting distorted face when the speaker faces forward.

on the noise produces more natural looking limbs. While, as mentioned previously predicting the denoised sample directly instead of predicting the noise shows good results in other work (EDGE [21] and MDM [20]), this direct prediction leads to some artifacts not shown in the noise prediction model. However, with these artifacts, the method still outperforms the other baseline methods, and, potentially with additional losses, this method shows to be a promising direction for improving on this work.

**User Study.** The metrics used to quantitatively measure the video generation aim to mimic human perception and mirror human quality assessment but leave room for improvement. As such, we conduct a user study to better validate the qualitative performance of our model. The study consists of 10 participants, who grade videos based on the quality of the generated gestures rather than the images. Specifically, we take 10 audios to generate videos for 5 different methods. The methods include the ground truth keypoints, DiffTED (our method), DiffTED but predicting keypoint position rather than the noise, the EAMM-based method [9], and the Speech2Gesture-based method

| Method | Naturalness | Smoothness | Synchrony |
|---|---|---|---|
| GT | 4.25 | 4.16 | 4.35 |
| EAMM [9] | 2.02 | 1.76 | 1.97 |
| S2G [5] | 2.45 | 2.31 | 2.30 |
| Position | 2.86 | 2.57 | 2.65 |
| Ours | **3.35** | **3.33** | **3.21** |

Table 3. User study results. The ratings on naturalness of gesture, smoothness of gesture and synchrony between speech and gesture are done on a scale of 1 to 5, where 5 is the best.

[5], with the order of these methods being shuffled for each audio. The participants are asked to grade the videos based on the smoothness of the gesture, the naturalness of the gesture, and the synchrony of the speech and gesture. Gradings are done on a scale of 1 to 5 where 5 is the best. Table 3 shows the results for the user study. Our method performed better than both baselines in all metrics, with only the ground truth performing better. The diffusion on the position rather than on the noise also performed better than both of the baselines.

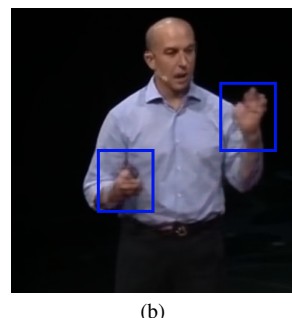

(a)                                    (b)

Figure 7. Images show examples of distorted hands showcasing the blurry artifacts that can occur, hands are highlighted in blue.

## 5. Limitations and Future Work

While DiffTED is able to create compelling videos from generated gestures, the gestures focus mainly on the body. While the head and the rest of the body are moved, the face often barely moves and does not always appear to be speaking. Additionally, there are some artifacts in the rendering process when side-views are used as source images. The diffusion model is able to create realistic gestures that look to the side and to a forward facing position, however, the inpainting network is unable to fill in the missing half of the body, this is most noticeable in the face as shown in the examples in Fig. 6. These types of issues can be mostly avoided by selecting front-facing views for the source images.

Additionally, the video rendering creates some blurry artifacts in the final images, this is mostly noticeable in the hands of the speaker. Figure 7 shows two examples of blurry, distorted hands. Because of occlusion of the fingers and the lack of keypoints specifically tracking the finger position, rendering hands proves to be a non-trivial problem.

For expanding on this work, we aim to incorporate a more robust face generation method to control the face and generate compelling talking faces. Additionally, adding an image refinement network to improve image quality and rectify the blurry artifacts is potentially a promising direction.

## 6. Conclusion

In this work, we present DiffTED, the first one-shot audio-driven video generation with diffusion-based co-speech gestures. We utilize the diffusion model to create coherent and diverse audio-driven gestures, represented as TPS keypoints. These TPS keypoints then drive the transformation of a single image to create realistic TED talk style videos. Our experiments show that a diffusion model can outperform EAMM and Speech2Gesture-based approaches in creating temporally consistent videos and realistic individual frames when utilizing the same one-shot image rendering method.

Our work is focused on producing TED talk style videos from a single image and a driving speech audio. The intended application of these style videos is to expand the ability for people to make presentation style videos in the same vein as TED talks. However, we have to recognize the potential for misuse and the ability for our work to enable the dissemination of disinformation. Proper use of this work will, we hope, enable educational talking videos in the style of TED talks and also enable the improvement of methods used to detect fake videos.

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
