# OpenReview forum: "DiffTED: One-shot Audio-driven TED Talk Video Generation with Diffusion-based Co-speech Gestures"
_thecvf.com/CVPR/2024/Workshop/HuMoGen — CVPR 2024 Workshop HuMoGen Submission_

### Official Review · Reviewer_zZ7G · 2024-03-30
**Original approach to gesture generation. The paper lacks in terms of clarity and missing crucial implementation details**

**Rating:** 3
**Confidence:** 3

**Review:**

Pros:
-  Unique approach for conditional gesture generation using key points, allowing straight-forward solution for video-rendering, unlike many other methods which does not cover the rendering part.
- Comprehensive evaluations, using various metrics demonstrating different qualitative and quantitative characteristics of the results. Also, a lot of effort was made to overcome the differences between the proposed solution and the others, and to make a fair comparison (to an extent).
- Interesting ablation study. I found it interesting that predicting the noise worked better than predicting key-points positions in most metrics and it lead to an interesting discussion.
- Qualitative results look good.

Cons:
- The paper is not self-contained. The proposed method strongly relies on an existing solution called TPS Motion Model, which is assumed to be known by the readers and not presented (not even in related work section). Furthermore, in section 2.1 the authors refer to specific components of TPS Motion Model, when the goal of referenced solution is not clear at this point.
- Missing details regarding the proposed solution, while less-relevant topics are overly detailed. In section 3.3, Diffusion-based TPS Keypoints Generation, the well-known diffusion process is told in details while the authors barely state any insights of their own or supply any information regarding their specific implementation. The authors didn't supply any information of how the condition was incorporated.
- Missing citations to support statements: L137-139, L139-141.
- Results seems a bit cherry-picked. In addition, on supplementary video, faces are blurred in claim that the solution does not deal with faces, while in the paper other solutions failure on faces are presented as disadvantage.
- Missing experiment details: Information about the test set which was used for the evaluation is missing, and also it's unclear how the audio and initial keypoints conditions were incorporated in the compared solution, which makes the comparison fairness uncertain.
- Missing related works: as mentioned before, TPS Motion Model is missing from the related work section, but also diffusion-based gesture generation solutions are missing, such as:
1. Anna Deichler, Shivam Mehta, Simon Alexanderson, and Jonas Beskow. 2023. Difusion-based co-speech gesture generation using joint text and audio representation. In Proceedings of the ACM International Conference on Multimodal Interaction (ICMI ’23). ACM.
2. YANG S., WU Z., LI M., ZHANG Z., HAO L., BAO
W., CHENG M., XIAO L.: Diffusestylegesture: Stylized audio-driven
co-speech gesture generation with diffusion models. arXiv preprint
arXiv:2305.04919 (2023)

Comments/Requests for final version (if accepted): Please add information about the condition representation and incorporation in your framework, add diffusion-based gesture generation solutions citations and also explain why you couldn't compare with these solutions.

The proposed method offers original end-to-end solution to the problem it aims to solve, and comprehensive evaluation of the results was conducted. At the same time, the paper lacks in terms of clarity, and crucial information about the method is missing. Considering both pros and cons, I've decided to rate this paper as borderline.

---

### Official Review · Reviewer_uVR1 · 2024-03-31
**A diffusion-based co-speech gesture video generation framework that enables the generalization to an unseen speaker with a single source image. I recommend acceptance.**

**Rating:** 4
**Confidence:** 5

**Review:**

Most of co-speech gesture video generation works require model fine-tuning on additional corresponding speaker data to generalize to a new, unseen speaker. The paper proposes a framework that supports a new speaker with a single source image. Specifically, the authors train a diffusion model to generate gesture sequences represented by general TPS keypoints. Then, a video renderer drives the source speaker image, conditioned on the generated gesture sequences. This setting is practical and has the potential to evolve into a useful product (The final video quality needs optimization).

**Pros**
* A setting that enables the generalization of a gesture model to an unseen speaker with a single source image.
* The paper explores a diffusion model to translate speech into 2D TPS keypoints, demonstrating its superior performance over LSTM-based and CNN-based models.

**Cons**
* Some blurry artifacts are present in the final video results (as discussed in Section 5). Employing a more advanced rendering module, such as Champ [1], may improve the visual quality.
* The lip movements appear to be poorly aligned with speech.

In summary, the one-shot setting is intriguing and could inspire subsequent research. The authors present a good method demonstrating the potential for constructing such a one-shot generation framework. I recommend acceptance.

---

### Meta-Review · Area_Chair_a3ZV · 2024-04-04

**Recommendation:** Accept

**Metareview:**

The paper addresses the task of creating a video with gestures from a single input image accompanied by speech input.

Pros:
* Predicting TPS points enables seamless video generation
* Comprehensive evaluation and ablation

Cons:
* Limited novelty
* The paper lacks clarity and is missing many details.
* The authors do not show results on the faces, while they criticize other methods on this matter.

The discussion about this paper was extensive, making it challenging to reach a decision. There was a notable concern regarding the writing quality. Ultimately, the decision was made to accept the paper with a firm recommendation to address the reviewers' feedback and rectify any identified issues.

**Guidance to authors:** Resolve concerns raised by the authors, particularly ones related to the clarity of writing.

---

### Decision · Program_Chairs · 2024-04-06

**Decision:**

Accept

**Comment:**

The paper will be published as part of the official CVPR workshop proceedings upon submission of the camera-ready version.